# New Green Tax Reforms: Ex-Ante Assessments for Spain

**Xavier Labandeira** [1,*]**, José M. Labeaga** [2] **and Xiral López-Otero** [2]

[1]   Rede, Universidade de Vigo, Facultade de CC.EE, Campus As Lagoas s/n, 36310 Vigo, Spain
[2]   Departamento de Teoría Económica y Economía Matemática, UNED, Senda del Rey 11, 28040 Madrid, Spain;
    jlabeaga@cee.uned.es (J.M.L.); xiral@outlook.es (X.L.-O.)
*   Correspondence: xavier@uvigo.es

**Abstract:** The great recession brought an increased need for public revenues and generated distributive concerns across many countries. This has led to a new generation of green tax reforms characterized by the use of markedly heterogeneous proposals that, overall, share a more flexible use of tax receipts adapted to the new economic environment. This article explores the possibilities of implementing this new generation of green tax reforms in Spain. It analyzes the impact of such reforms on energy demand, emissions, public revenues and income distribution from taxing various energy-related environmental damages and by considering two alternative uses for the tax receipts: fiscal consolidation and funding the costs of renewable-energy support schemes.

**Keywords:** taxes; renewables; energy; negative externalities; new generation green tax reforms; climate change mitigation; Spain

---

## 1. Introduction

Energy taxes have traditionally been used for revenue purposes, as a consequence of the low price elasticity of energy products in general (see [1]), which allows taxes on these products to be a high and stable source of collection. Furthermore, since the early 1990s, the incorporation of environmental objectives has gradually become common practice in most countries [2] and, given that many of the environmental problems we currently face are directly or indirectly related to energy, especially in the case of climate change (see [3–5]), there has been an increase in the use of energy taxes with environmental purposes. In some cases, the introduction of these taxes took place within a broader fiscal reform package, the so-called green tax reforms, using the revenues generated by the new environmental taxes to reduce other, more distorting, taxes or with other purposes (see Section 2).

At present, however, energy taxes are generally below their optimal environmental level in most countries, with hardly any substantial changes in recent years [6]. Given the serious environmental problems associated with energy, an increase in the use of energy-environmental taxation is highly recommended. At European Union (EU) level, although successive attempts to increase energy taxation (see [7,8]) were blocked by different countries due to the unanimity fiscal rule, the European Commission recently published a communication demanding the lift of unanimity for fiscal legislation on energy and the environment [9], which could facilitate the adoption of a new directive in this area (to substitute the outdated EU fiscal rules from the early years of the century).

In Spain, despite favorable academic evidence (see Section 2), environmental-energy taxation has played a limited role [10] by only incorporating environmental grounds within the tax system in a reduced or indirect manner, sometimes even incentivizing negative environmental behavior. As a result, energy taxation in Spain is below that of most EU countries, as shown in Table 1. Indeed, Spain ranks last in the EU in terms of energy and environmental tax revenues (environmental taxes in

Spain in 2017 represented 5.4% of the tax revenue and 1.8% of GDP, as compared to the respective 6.1% and 2.4% in EU-28. Taxes on energy accounted for 4.5% of the revenue and 1.5% of GDP in Spain, as compared to the respective 4.7% and 1.8% in the EU-28 [11]).

**Table 1.** Energy taxes in several European countries (% on energy prices), 2018.

| Country | Electricity (Households) | Electricity (Industrial) | Natural Gas (Households) | Natural Gas (Industrial) | Automotive Diesel (Non-Commercial) | Automotive Diesel (Commercial) | Unleaded Gasoline (95 RON) |
|---|---|---|---|---|---|---|---|
| France | 36.20% | 22.08% | 27.04% | 16.23% | 59.41% | 51.29% | 62.43% |
| Germany | 53.83% | 49.10% | 24.38% | 15.67% | 52.40% | 42.61% | 60.64% |
| Italy | 32.82% | 34.83% | 35.80% | 11.93% | 59.87% | 51.01% | 63.58% |
| Spain | 21.39% | 4.88% | 20.25% | 2.16% | 47.65% | 36.66% | 52.89% |
| UK | 4.75% | 3.82% | 4.76% | 3.52% | 61.82% | 54.22% | 63.13% |
| EU-23 [1] | 31.04% | 21.43% | 23.70% | 10.59% | 55.00% | 45.49% | 60.22% |

[1] Weighted average by population of the 23 countries of the EU belonging to the OECD. Source: [12].

At a time when, in contrast to international and European reduction objectives, Spanish $CO_2$ emissions are only undergoing small reductions (−3.2% in 2018 with respect to 2017 [13], which had seen a 7.4% annual increase [14]), and in a situation where public accounts have yet to recover from the economic crisis (in Spain, public deficit stood at 2.5% of GDP [15] in 2018, and the share of public revenues in GDP is still 5.1% lower than it was before the beginning of the economic crisis [16]), there are numerous reasons and great scope to increase energy and environmental taxes.

In this context, this article aims to analyze the effects of a third-generation green tax reform in Spain by introducing environmental taxes on the main energy products and two alternative uses for tax revenues: fiscal consolidation and the financing of the (significant) cost of supporting renewable energy. The results show these reforms are capable of generating additional revenue while reducing energy consumption and $CO_2$ emissions with limited and generally progressive distributive impacts. This work therefore contributes to expand the short supply of international academic literature on third-generation green tax reforms, practically nonexistent in the case of Spain.

The paper is structured in 6 sections, including this introduction. The second section describes the theoretical context of the article, the third section presents the data and the methodology used to prepare the study, while the fourth section shows the results obtained. Finally, the paper incorporates a section on the analysis of the results and implications, and a final concluding section.

## 2. Theoretical Context

Although the possibility of obtaining additional fiscal benefits from environmental taxes were first mentioned in the 1960s [17], theoretical literature on this matter began to develop thirty years later when the potentially high and stable revenues associated with carbon energy taxes made these taxes suitable to lead tax reform processes [18]. Thus, the "double dividend" theory [18] indicates that a further benefit to welfare could be achieved if, in addition to the environmental benefits (static, cost-effective attainment of objectives, and dynamic, technology-related) obtained by introducing an environmental tax, the revenue aimed to reduce the size of other more distorting taxes (i.e., if a "green tax reform" was implemented). Initially, the vision of welfare gains from environmental taxation, the so-called "strong" double dividend [19], was too optimistic given that the effect on the non-environmental welfare was assumed to be either null or positive. However, the work of [20] showed that environmental taxes generate additional efficiency costs by distorting the markets for goods and factors; thereby, environmental taxes also increase pre-existing distortions. A broad theoretical literature on double dividend resulted from these studies. It incorporated issues like intermediate inputs [21], capital mobility [22], involuntary unemployment [23,24], unemployment benefits [25], tax-favored consumer goods [26], oligopoly [27], fixed production factor [28], black economy [29], or tax evasion [30].

At any rate, the general consensus accepts the presence of a second "weak" dividend, defined as the efficiency gain derived from allocating the revenue obtained with environmental taxation to allow

for the reduction of other more distorting taxes (when compared to other alternatives). Subsequent to the theoretical advances, a rich empirical literature emerged as theoretical literature on the double dividend of environmental taxation developed, focusing on the impacts of green tax reforms generally through ex-ante simulations ([31–36] provide summaries of the methodologies and results). This literature generally points out that green tax reforms allow significant reductions in pollution at a limited economic cost [34,37] because recycling tax revenue helps mitigate the negative macroeconomic effects of environmental taxation [36].

In this context, some countries began to implement this tax reform model in practice. In general, the literature (see [33–35,38,39]) distinguishes between two generations of green tax reforms that follow the foundations of the double dividend theory: the use of environmental tax revenue to reduce other conventionally distorting taxes within a context of full revenue substitution. The first generation began in the early 90s of the last century in Scandinavia. It used strong environmental taxes closely related to the energy sector and recycled the revenue obtained to reduce personal income tax and corporate tax (Sweden, 1991, Norway, 1992, The Netherlands, 1992). The second generation includes the solutions applied at the turn of the century, basically employing environmental tax revenues to reduce social security contributions and simultaneously applying compensatory measures for the most affected groups or sectors/industries (United Kingdom, 1996, Finland, 1998, Germany, 1999, Estonia, 2006, Czech Republic, 2008).

However, among other reasons, the great recession and the growing need for public revenues across many countries; a more intense promotion of renewable energies and energy efficiency; and the increase in distributional and competitiveness concerns all led to a third generation of green tax reforms [35] that encompassed a set of rather heterogeneous proposals that moved away from the standard double dividend reasoning and used the revenue more flexibly to adjust to the new socio-economic situation. Examples of countries that have implemented a third generation green tax reform are Switzerland [40], which introduced a tax on carbon dioxide ($CO_2$) emissions in 2008 and partly used its proceeds to promote energy efficiency in buildings and compensatory measures for affected households and businesses; Ireland [41], which established a carbon tax at the end of 2009 and allocated revenue to fiscal consolidation; Slovenia [42], which applied a tax on energy consumption since 2010 and entirely devotes its revenue to financing energy efficiency programs; Japan [43], which in 2012 approved a tax on $CO_2$ emissions and employs its revenue to climate change mitigation; and the Netherlands [44], which in 2013 introduced a surcharge on energy taxation and uses this revenue to fund renewable production.

Nevertheless, while the double dividend literature provides a basis and allows for the evaluation of first- and second-generation green tax reforms, academic evidence on green third-generation fiscal reforms is scarce. Exceptionally, we can mention [45,46], which incorporate distributive aspects in the analysis of a green tax reforms; [47], which analyzes the effects of introducing a tax to finance a renewable subsidy; [48], which studies the distributive effects of green tax reforms in the presence of heterogeneous households; [49], which considers introducing a tax on electricity and allocating this revenue to R&D in $CO_2$ abatement; [50], which contemplates the introduction of a carbon tax whose revenues are transferred to poor households; [51], which studies the effect of allocating environmental tax revenue to increasing public spending; [52], which evaluates different schemes to transfer carbon tax revenue to households; [53], which studies different recycling alternatives (support for renewables, promotion of energy efficiency and distributive compensations); and [54], which analyzes the distributional impact of $CO_2$ tax with hybrid recycling through tax reductions and lump sum compensations to low-income households.

In the Spanish case, the empirical literature on the effects of green tax reforms is sparse. It focuses on first- and second-generation reforms, within a neutral tax revenue setting and compensations on social security levies [55–65], VAT [66], or other alternatives [67–69]. In general, and in line with the international empirical literature, results show that these reforms could reduce energy consumption and emissions without a significant macroeconomic impact. In fact, they are generally positive in terms

of employment and welfare in the case of social security compensations. In terms of distributive effects, they are generally slightly regressive, but less so than those observed in other developed countries.

## 3. Methodology and Material

### 3.1. Data

Our study takes into account the main energy products consumed by Spanish households (electricity, natural gas, gasoil A and gasoline 95) in 2016. The data on energy consumption were obtained from [70] (electricity) and [71] (natural gas, gasoil A and gasoline 95) (residential energy consumption has been calculated from the total consumption data using information from [72] (electricity and natural gas) and [73] (diesel A and gasoline 95). It is assumed that the remaining energy consumption has industrial and commercial origin. The Canary Islands, Ceuta and Melilla are excluded from the analysis because they do not apply the national tax on hydrocarbons. However, the national tax on electricity is applied in these areas: hence our consideration for revenue calculations, although they are not considered in the distributive analysis so that comparisons between the different reforms come from the same sample), while the prices and taxes applied to these products were obtained from [74]. The tax burden on electricity goes beyond traditional taxes (VAT and special taxes), including charges to finance different public policies (thus, the electricity charges devoted to finance public policies in 2015 represented an extremely important part of the final price of these products (28.8% of the final residential price and 24.8% of the final industrial price [75]). Outstanding among these charges were those employed to finance the cost of renewables, cogeneration and waste. In 2016, 19.4% of the average final price of electricity went for this purpose), so we have used information from [70,75] to break down the different tax charges supported by this product (see Table 2).

**Table 2.** Prices and taxes on energy products (€/MWh). 2016.

| Energy Product | Type of Consumer | Prices and Taxes on Energy Products (€/MWh). 2016 | Excise Tax | VAT | Costs of Support to Renewable, Cogeneration and Waste | Other Charges | Final Price |
|---|---|---|---|---|---|---|---|
| Electricity | Residential | 119.21 | 5.11% [2] | 21% [1] | 51.37 | 17.57 | 239.3 |
| Electricity | Industrial | 73.46 | 5.11% [2] | - | 14.34 | 11.62 | 104.5 |
| Natural gas | Residential | 63.91 | 2.34 | 21% [1] | - | - | 80.16 |
| Natural gas | Industrial | 23.35 | 0.54 | - | - | - | 23.89 |
| Diesel | Residential | 47.73 | 37.37 | 21% [1] | - | - | 102.97 |
| Diesel | Industrial | 47.73 | 37.37 | - | - | - | 85.1 |
| Gasoline | Residential | 53.82 | 50.85 | 21% [1] | - | - | 126.65 |
| Gasoline | Industrial | - | - | - | - | - | - |

[1] Ad valorem tax on the price before taxes and other charges. [2] Ad valorem tax on the VAT base. Source: [70,74,75] and the authors.

We employ the price elasticities calculated for Spain (see Table 3) in a meta-analysis of the literature [76] (Ref. [76] carried out a meta-analysis of the estimates in the literature of the price-elasticities of demand for energy products in Spain, which constitutes an adaptation to Spain of [1], in which a meta-analysis of these elasticities is carried out at a global level. Short-term elasticities are considered and cross-elasticities are not taken into account, since substitution effects would require some investment and therefore they would correspond to long-run effects, while our estimates correspond to short-run effects) to calculate the impact on consumption of the price change resulting from the reforms under study, while considering the emission factors of [77] carbon dioxide ($CO_2$), [78] nitrogen and sulfur oxides of liquid fuels (NOx and $SO_2$ respectively), [79] (NOx of natural gas) and [80] ($SO_2$ of natural gas) to transform the energy consumed in emissions.

**Table 3.** Energy price and demand elasticities.

| Product | Elasticity |
|---|---|
| Electricity | −0.203 |
| Natural gas | −0.242 |
| Diesel | −0.201 |
| Gasoline | −0.253 |

Note: We consider the same elasticities for residential and industrial consumers, since the meta-analysis presents no statistically significant differences between them. Source: [76].

Finally, we use the 2016 microdata of the Spanish Family Budget Survey (EPF), prepared by the National Institute of Statistics (INE), to carry out the distributive analysis. Our available observations for 22,011 households are fully representative of the Spanish population through the use of the grossing-up factor (the grossing-up factor indicates the total population represented by each household in the sample). We consider total household expenditure as the income variable, calculate the impact of the reforms on the total expenditure of each household (we use the new prices of energy goods resulting from the reform to calculate the new consumption (based on the price elasticities in Table 3) and the expenditure of each household on the different energy products to evaluate the impact of the reform on the total expenditure of each household), and apply the grossing-up factor to this impact. This permits us to calculate the average effect per decile of income. Figure 1 shows that in 2016, the share of electricity in total spending decreased as the level of income increased, thereby causing taxes on this energy product to have the most regressive impact. On the other hand, the share of diesel and gasoline in total spending increased up to the eighth decile and decreased in the last two deciles. Finally, natural gas has a lower share in household spending, with a similar percentage of expenditure in all deciles (except for the last, which is slightly lower).

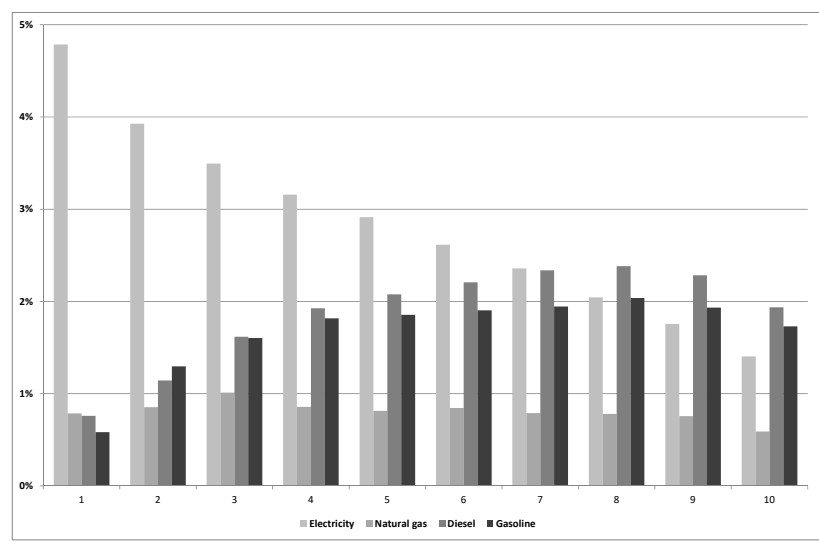

**Figure 1.** Percentage of expenditure on the different energy products per income decile, 2016. Source: EPF and the authors.

*3.2. Considered Reforms*

We contemplate four scenarios and calculate the impact of each of them on prices, demand and $CO_2$ emissions for the different energy products in both residential and industrial sectors. Using the prices and consumption resulting from the reform, we then calculate the revenue impact for both sectors. Finally, we use the microdata of the EPF to assess the distributive impact of each reform on Spanish households. To this end, we calculate the impact of the variation in the contemplated energy prices of on the consumption of each household, based on the elasticities in Table 3. With the

new prices and consumption, the effect of the reform is calculated on the total expenditure of each household and, taking into account the population elevation factor, the average effect per income decile is calculated. The scenarios employed in this paper update the results of [81], incorporating new simulations, disaggregating residential and industrial sectors, and implementing methodological improvements on [82].

### 3.2.1. Scenario 1. Increased Excise Taxes on Energy

Bearing in mind the importance of coordinating tax reforms at EU level in order to prevent member-state actions from weakening the effectiveness of European tax systems and affecting the functioning of the internal market [83], the first scenario contemplates two reforms that facilitate the convergence of Spanish energy-environmental taxation with the fiscal situation of neighboring countries. In this sense, in 2011, the European Commission presented a proposal for a directive to simultaneously tax the energy content and implicit $CO_2$ emissions of energy products. Thus, it defined a minimum level and structured tax rates in two sections, one based on energy content (for revenue purposes and energy security) and another that was based on the $CO_2$ content and linked to the European Emissions Trading System (EU ETS) [8]. Even though the proposal had to be abandoned a few years later due to the opposition of certain member states, in Scenario 1 we analyze the effects of introducing the minimums established in this proposal (see Table 4) in Spain. As an alternative to these minimum rates and taking into account Spain's low energy tax levels as compared to its neighboring countries (Table 1), we simulate the effects of introducing in Spain the weighted average of the energy taxes levied in four large European countries: Germany, France, Italy and the United Kingdom (Table 5). The additional revenue derived from these reforms would be destined to fiscal consolidation, i.e., to increasing government revenues.

**Table 4.** Minimums for 2018 of the 2011 Directive Proposal.

| Energy Product | Emissions (€/$CO_2$ ton) | Energy Consumption (€/GJ) | Tax Rate |
| --- | --- | --- | --- |
| Electricity | 0 | 0.15 | 0.540 €/MWh |
| Natural gas | 20 | 0.15 | 4.579 €/MWh |
| Diesel | 20 | 9.6 | 0.397 €/L |
| Gasoline | 20 | 6.6 | 0.353 €/L |

Source: [8,84] and the authors.

**Table 5.** Energy excise taxes applied in the main European countries, 2016.

| Energy Product | France | Germany | Italy | United Kingdom | Weighted Average |
| --- | --- | --- | --- | --- | --- |
| Electricity (residential) (€/MWh) | 34.86 | 110.70 | 69.00 | - | 56.73 |
| Electricity (industrial) (€/MWh) | 24.79 | 61.40 | 70.70 | 4.27 | 40.93 |
| Natural gas (residential) (€/MWh) | 5.58 | 5.50 | 15.22 | - | 6.34 |
| Natural gas (industrial) (€/MWh) | 3.94 | 4.03 | 4.46 | 0.77 | 3.32 |
| Diesel (€/L) | 0.511 | 0.470 | 0.617 | 0.708 | 0.569 |
| Gasoline (€/L) | 0.648 | 0.655 | 0.728 | 0.708 | 0.682 |

Source: [74] and the authors.

### 3.2.2. Scenario 2. New Taxes on Emissions

The second scenario considers the introduction of taxes on emissions associated with the consumption of energy products. That is to say, the introduction of taxes on $CO_2$ emissions, the main

greenhouse gas, as well as the emissions of $SO_2$ and NOx, the main causes of acid rain that also represent an important hazard to human health (see [85]).

Thus, we simulate the introduction of a tax on $CO_2$ emissions in sectors that are not subject to the EU ETS—principally the transport, residential and commercial sectors—that contemplates the varying carbon content of energy products (greenhouse gas emissions from the transport sector in Spain in 2017 represented 26% of the total, while the residential, commercial and institutional sectors accounted for 8% of emissions. On the other hand, the sectors included in the EU ETS generated 40% of the total greenhouse gas emissions, almost half of these emissions (20% of the total) coming from the generation of electricity [86]). Two tax levels are considered. €10/t$CO_2$ and €30/t$CO_2$. The first is similar to the tax rate that would allow for a significant cost-effective reduction of $CO_2$ emissions in Spain [87]. Alternatively, we simulate a tax of €30/t$CO_2$, considering the cost of the externalities associated with $CO_2$ emissions [88], as well as several opinions in the academic literature. We assume that the price of electricity is only affected in the second case (€30 /t$CO_2$), obtaining that impact from [82,89], and unaffected in the first case (€10/t$CO_2$) because the electricity sector is included in the EU ETS (in addition, for electricity we consider the revenue derived from the increase in the $CO_2$ price in the EU ETS as $CO_2$ tax revenue, assuming that the public sector would obtain them through auctions).

Likewise, we also consider introducing a tax on NOx and $SO_2$. Although the usual estimates [88] consider externalities of up to €14,000/t for NOx and €18,000/t for $SO_2$, the actual tax rate applied on these products is much lower. In this context, we have chosen to use lower figures that are closer to what is actually applied. We thus use a tax rate of €1000/t as the lower threshold and a tax rate of €2000/t as the upper threshold.

In this second scenario, we consider two reforms. The first consists of introducing a €10/t tax on $CO_2$ and a €1000/t tax on NOx and $SO_2$. The second uses the same taxes with higher tax rates (€30/t$CO_2$ and €2000/t of NOx and $SO_2$). As in Scenario 1, the additional revenues obtained with these reforms are devoted to fiscal consolidation.

### 3.2.3. Scenario 3. Eliminating the Electricity Tariff from the Support Costs for Renewable, Cogeneration and Waste

Promotion of renewable energy is one of the most relevant climate mitigation policies. Although this strategy should involve all energy sectors, it is the electricity sector that has historically made the greatest effort to promote renewable energy given the lower costs of its alternatives. As a result of transferring these differentiated efforts to final prices, Spanish electricity consumers are currently supporting the greatest part of the financing effort to promote renewable energies. In 2015, electricity accounted for 26% of final energy consumption, but it supported 88% of the costs of promoting renewables in Spain [90]. Therefore, the financing mechanism for renewables in Spain discourages the electrification of the economy and implicitly encourages the consumption of fossil fuels.

In this scenario, we simulate the elimination of charges destined to renewable energies, cogeneration and waste from the electricity tariff. We study two alternatives to obtain the necessary revenue to finance this public policy. To this end, we consider introducing a tax on energy products that generates a revenue equivalent to that collected from the existing electricity charges so that costs are distributed among the four energy products proportionately to their consumption [91], and inversely proportionally to its price elasticity to minimize distortions in the economy (Ramsey) [92]. This way the reform is revenue neutral in both cases. Table 6 shows the distribution based on these two criteria (it is assumed that with the new tax each sector (residential/industrial) provides similar funds to those obtained through the electricity charges to finance renewables).

**Table 6.** Distribution by energy product of the amount to be financed in Scenario 3.

| Product | Consumption | | Ramsey | |
|---|---|---|---|---|
| | **Residential** | **Industrial** | **Residential** | **Industrial** |
| Electricity | 20.76% | 36.67% | 27.39% | 35.10% |
| Natural gas | 16.48% | 42.55% | 22.97% | 29.45% |
| Diesel | 48.43% | 20.79% | 27.66% | 35.45% |
| Gasoline | 14.33% | - | 21.98% | - |

Source: The authors.

### 3.2.4. Scenario 4. Taxes on Emissions and the Financing of Renewables

This scenario is a combination of the previous scenarios with the suppression of costs supporting renewable energies, cogeneration and waste in parallel to the introduction of taxes on emissions. The first simulation analyzes the effects of introducing a tax of €30/t on $CO_2$ emissions and another tax of €2000/t on NOx and $SO_2$ emissions, while eliminating the cost of renewables, cogeneration and waste from the electricity bill. Given that the tax revenue fails to cover the entire cost of renewables, the second simulation considers the tax rates on emissions that would be required to ensure that the reform is revenue neutral.

## 4. Results

### 4.1. Scenario 1. Increase in the Excise Taxes on Energy

#### 4.1.1. Reform 1A. Minimums for 2018 Directive Proposal. Fiscal Consolidation

First, we simulate a reform that consists of increasing the excise taxes on energy products up to the minimum levels for 2018 of [8] and allocating the additional revenue generated to fiscal consolidation. Given that the minimum levels (susceptible to an increase by each EU member state) established [8] for electricity and gasoline are below the excise rates currently applied in Spain, this first simulation makes no modification on the taxes levied on these products and only increases the excises on natural gas and diesel.

The impacts of this reform are rather negligible (Table 7). It would lead to an increase in the final prices of the affected products and thus reduce the consumption of natural gas and diesel by 1.36% and 0.70%, respectively, while the $CO_2$ emissions derived from energy products would fall by 0.55%. The reform allows for a 6% increase in tax revenue associated with energy products (around 1700 million euros) that would mainly come from excise taxation. The additional revenue, by energy product, would mainly come from natural gas (Table 8).

**Table 7.** Reform 1A. Effects on energy products.

| Product | Price Variation | | Consumption Variation | | |
|---|---|---|---|---|---|
| | **Residential** | **Industrial** | **Residential** | **Industrial** | **Total** |
| Electricity | - | - | - | - | - |
| Natural gas | 3.38% | 6.27% | −0.82% | −1.52% | −1.36% |
| Diesel | 3.46% | 3.46% | −0.70% | −0.70% | −0.70% |
| Gasoline | - | - | - | - | - |

Source: The authors.

**Table 8.** Reform 1A. Revenue change. Millions of euros.

|  |  | Excise Tax | VAT | Total | Total (%) |
|---|---|---|---|---|---|
| Electricity | Residential | - | - | - | - |
|  | Industrial | - | - | - | - |
|  | Total | - | - | - | - |
| Natural gas | Residential | 125.81 | 20.15 | 145.96 | 15.72 |
|  | Industrial | 766.88 | - | 766.88 | 735.10 |
|  | Total | 892.69 | 20.15 | 912.84 | 88.36 |
| Diesel | Residential | 447.70 | 82.30 | 530.01 | 5.71 |
|  | Industrial | 251.52 | - | 251.52 | 7.13 |
|  | Total | 699.23 | 82.30 | 781.53 | 6.10 |
| Gasoline | Residential | - | - | - | - |
|  | Industrial | - | - | - | - |
|  | Total | - | - | - | - |
| **Total** | **Residential** | **573.52** | **102.45** | **675.97** | **3.16** |
|  | **Industrial** | **1018.40** | **-** | **1018.40** | **14.57** |
|  | **Total** | **1591.92** | **102.45** | **1694.37** | **5.97** |

Note: The last column depicts the change in revenue with respect to the baseline. Source: The authors.

The distributive effects of the reform (Figure 2) are determined mainly by its impact on diesel. Consequent to the distribution of diesel spending by income deciles, the percentage reduction in the level of household income is larger as the level of income increases up to the eighth decile. From there onward, the impact of the reform will diminish because the richest households spend a smaller proportion of their income on diesel (see Figure 1). The general effects of the reform are small, but they are progressive and have a greater impact on households with higher income. The slight reduction of the Gini Index (0.01%) reflects this.

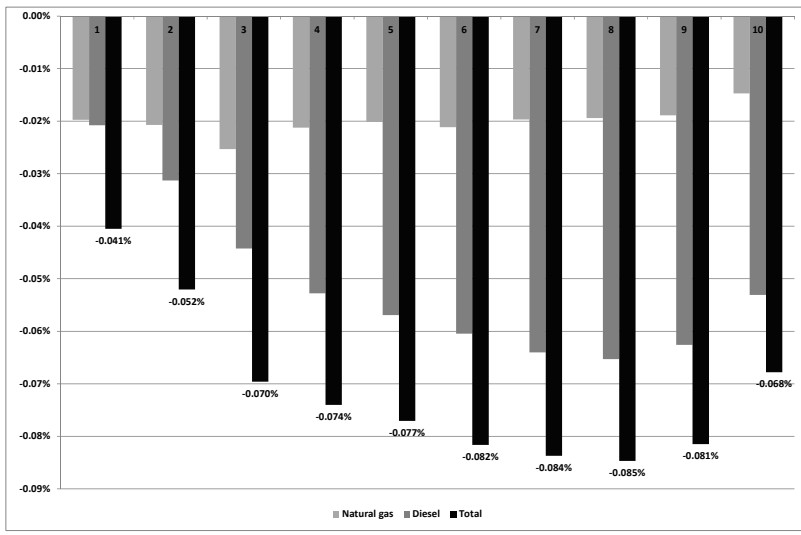

**Figure 2.** Reform 1A. Distributive impact by income deciles (%). Source: The authors.

### 4.1.2. Reform 1B. Weighted Average of the Main EU Countries. Fiscal Consolidation

The impact of the reform would be much greater if excises were raised to the level of the weighted average of Germany, France, Italy and the United Kingdom, instead of increasing the excise taxes on energy products up to the minimum levels of the directive. This alternative would produce a significant increase in the price of all energy products (Table 9) and cause a significant reduction in the aggregate consumption of energy products (−4.2%) and associated $CO_2$ emissions (−4.6%).

**Table 9.** Reform 1B. Effects on energy products.

| Energy Product | Price Variation | | Consumption Variation | | |
|---|---|---|---|---|---|
| | Residential | Industrial | Residential | Industrial | Total |
| Electricity | 23.82% | 34.30% | −4.84% | −6.96% | −6.32% |
| Natural gas | 6.04% | 4.31% | −1.46% | −1.04% | −1.14% |
| Diesel | 23.99% | 23.99% | −4.82% | −4.82% | −4.82% |
| Gasoline | 23.13% | - | −5.85% | - | −5.85% |

Source: The authors.

This reform would allow for a very significant increase in the tax revenue associated with energy products (Table 10). It would generate nearly 16000 million additional euros that would mainly come from the new excise taxes, although the revenue derived from VAT would also increase substantially. By energy product, the additional revenue would mainly come from electricity (about 8900 million euros) and diesel (5166 million euros), while the contribution of natural gas and gasoline would be smaller.

**Table 10.** Reform 1B. Revenue change. Millions of euros.

| | | Excise Tax | VAT | Renewables | Total | Total (%) |
|---|---|---|---|---|---|---|
| | Residential | 3194.66 | 533.30 | −188.17 | 3539.79 | 46.73 |
| Electricity | Industrial | 5493.59 | - | −174.85 | 318.75 | 158.43 |
| | Total | 8688.25 | 533.30 | −363.02 | 8858.54 | 81.04 |
| | Residential | 223.30 | 35.69 | - | 258.98 | 27.88 |
| Natural gas | Industrial | 530.37 | - | - | 530.37 | 508.40 |
| | Total | 753.67 | 35.69 | - | 789.36 | 76.40 |
| | Residential | 2961.56 | 540.74 | - | 3502.30 | 37.73 |
| Diesel | Industrial | 1663.82 | - | - | 1663.82 | 47.17 |
| | Total | 4625.38 | 540.74 | - | 5166.12 | 40.33 |
| | Residential | 985.43 | 174.05 | - | 1159.48 | 32.02 |
| Gasoline | Industrial | - | - | | - | - |
| | Total | 985.43 | 174.05 | - | 1159.48 | 32.02 |
| | **Residential** | **7364.95** | **1283.78** | **−188.17** | **8460.55** | **39.52** |
| **Total** | **Industrial** | **7687.79** | **-** | **−174.85** | **7512.94** | **107.50** |
| | **Total** | **15052.73** | **1283.78** | **−363.02** | **15973.49** | **56.25** |

Note: The last column depicts the change in revenue with respect to the baseline. Source: the authors.

The impact of the reform at the household level would be much greater than in Reform 1A due to its effects on electricity and diesel prices (see Figure 3). The percentage reduction in the level of income would increase until the fourth decile and decrease thereafter, thus indicating a regressive impact. The 0.18% increase of the Gini index relative to the baseline confirms its regressivity.

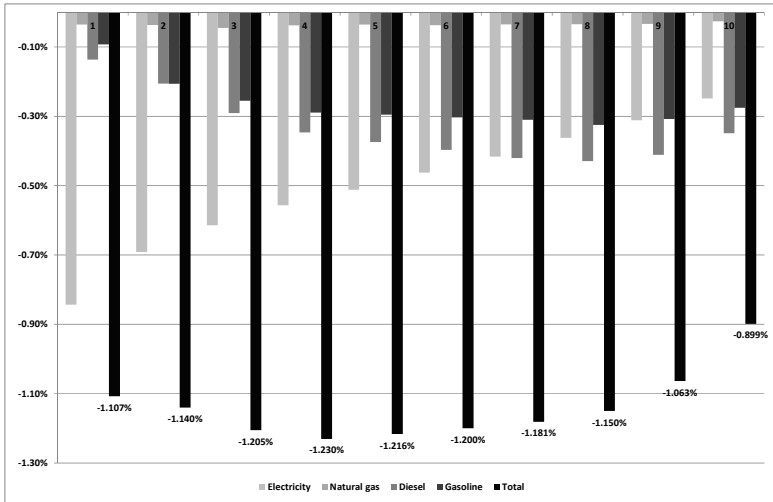

**Figure 3.** Reform 1B. Distributive impact by income deciles (%). Source: The authors.

*4.2. Scenario 2. New Taxes on Emissions*

4.2.1. Reform 2A. Taxes on Emissions for $CO_2$ (€10/t), $SO_2$ (€1000/t) and NOx (€1000/t). Fiscal Consolidation

On the one hand, a tax on $CO_2$ emissions of €10/t is introduced in sectors that are not subject to the EU ETS. This has no effect on the price of electricity because the electricity sector is part of this system (we assume that the price of $CO_2$ in the EU ETS is €10/t, as this was roughly the case in the simulation period), while it does on the price of all the other energy products. Taking the emission factors of each energy product (see Section 3.1) into account, we may see that introducing this tax would imply an additional charge of €1.82/MWh for natural gas, €0.025/L for diesel and €0.022/L for gasoline. Additionally, the reform includes a tax on NOx and $SO_2$ emissions of €1000/t, which translates into an additional €0.43/MWh for electricity, €0.18/MWh for natural gas, €0.011/L for diesel and €0.006/L for gasoline (in the case of electricity, given that NOx and $SO_2$ emissions depend on the generation mix, the equivalent charge is obtained from [82,89]) when the emission factors are introduced.

As a result, (Table 11), the price of energy products increases slightly (between 0.2–4.3%). This leads to small reductions in the consumption of these products (between 0.07% and 0.85%) and $CO_2$ emissions (0.53%). In this context, the tax increase is akin to that of Reform 1A (6.1%) and mainly comes from the new tax on $CO_2$ emissions (around 1200 million euros), while the tax on NOx (mainly) and $SO_2$ emissions would generate a revenue of about 460 million euros and the VAT revenue would increase slightly and compensate for small reductions the receipts from energy excise taxation and other charges. By energy product, the additional revenue would mainly come from diesel (around 960 million euros) and natural gas (approximately 510 million euros) (Table 12).

**Table 11.** Reform 2A. Effects on energy products.

| Energy Product | Price Variation | | Consumption Variation | | |
|---|---|---|---|---|---|
| | Residential | Industrial | Residential | Industrial | Total |
| Electricity | 0.22% | 0.42% | −0.04% | −0.08% | −0.07% |
| Natural Gas | 3.02% | 3.10% | −0.73% | −0.75% | −0.75% |
| Diesel | 4.25% | 4.25% | −0.85% | −0.85% | −0.85% |
| Gasoline | 2.99% | - | −0.76% | - | −0.76% |

Source: The authors.

**Table 12.** Reform 2A. Revenue changes. Millions of euros.

|  |  | Excise Tax | VAT | $CO_2$ Tax | NOx/SO2 Tax | Renewables | Total | Total (%) |
|---|---|---|---|---|---|---|---|---|
| Electricity | Residential | −0.31 | 5.23 | - | 31.28 | −1.74 | 34.47 | 0.46 |
|  | Industrial | −0.71 | - | - | 72.30 | −2.12 | 69.46 | 2.07 |
|  | Total | −1.02 | 5.23 | - | 103.58 | −3.86 | 103.93 | 0.95 |
| Natural gas | Residential | −0.98 | 18.01 | 103.19 | 10.21 | - | 130.44 | 14.04 |
|  | Industrial | −0.78 | - | 348.77 | 34.51 | - | 382.50 | 366.65 |
|  | Total | −1.76 | 18.01 | 451.96 | 44.72 | - | 512.93 | 49.65 |
| Diesel | Residential | −53.69 | 100.96 | 423.43 | 179.59 | - | 650.29 | 7.01 |
|  | Industrial | −30.16 | - | 237.89 | 100.89 | - | 308.62 | 8.75 |
|  | Total | −83.85 | 100.96 | 661.32 | 280.48 | - | 958.91 | 7.49 |
| Gasoline | Residential | −19.10 | 24.13 | 119.26 | 34.93 | - | 159.22 | 4.40 |
|  | Industrial | - | - | - | - | - | - | - |
|  | Total | −19.10 | 24.13 | 119.26 | 34.93 | - | 159.22 | 4.40 |
| **Total** | **Residential** | **−74.07** | **148.33** | **645.88** | **256.01** | **−1.74** | **974.42** | **4.55** |
|  | **Industrial** | **−31.66** | **-** | **586.65** | **207.70** | **−2.12** | **760.58** | **10.88** |
|  | **Total** | **−105.73** | **148.33** | **1232.53** | **463.71** | **−3.86** | **1735.00** | **6.11** |

Note: The last column depicts the change in revenue with respect to the baseline. Source: The authors.

The distributive effects fundamentally derive from the increase in the price of diesel and, although small, they are slightly progressive (Figure 4). All deciles show reduced income levels, with a greater percentage of reduction as the level of income increases up to the eighth decile; while the Gini index slightly decreases (−0.01%).

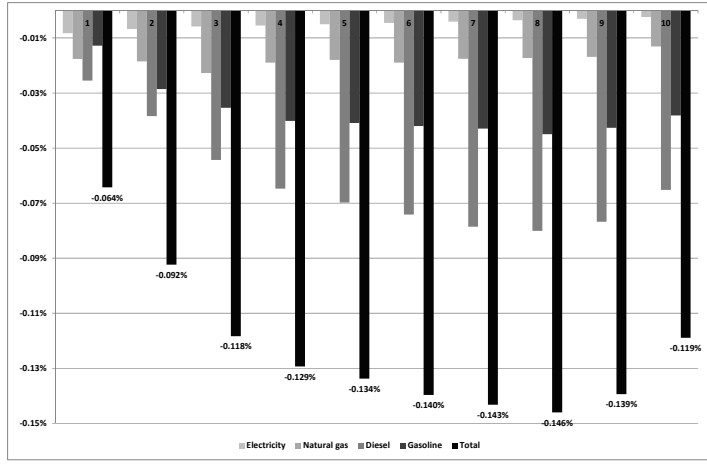

**Figure 4.** Reform 2A. Distributive impact by income deciles (%). Source: The authors.

4.2.2. Reform 2B. Taxes on $CO_2$ Emissions (€30/t), $SO_2$ (€2000/t) and NOx (€2000/t). Fiscal Consolidation

In this case, we introduce the same taxes as in the previous simulation with higher tax rates. On the one hand, $CO_2$ emissions are taxed at a rate of €30/t, assuming that the price of these emissions in the EU ETS also increases to that level (as is the case at the moment of writing). Under these circumstances, additional tax rates of €5.46/MWh for natural gas, €0.075/L for diesel and €0.066/L for gasoline would be implemented, while from [82,89] we obtain the increase in the (pre-tax) price of electricity of €2.61/MWh. On the other hand, the new tax of €2000/t on NOx and $SO_2$ emissions would translate into an additional charge of €1.43/MWh for electricity, €0.36/MWh for natural gas, €0.021/L for diesel and €0.013/L for gasoline.

The resulting impacts of the reform would be substantial (Table 13). Thus, the prices of energy products would increase between 2.1–11.5%, causing consumption reductions between 0.7% and 2.3% and a $CO_2$ emission reduction of 1.7%. In terms of revenue, this reform could generate 5500 million euros (an increase of 19.2%), mainly from the tax on $CO_2$ (4261.8 million euros. As explained above,

this includes revenues derived from the increase in the price of $CO_2$ in the EU ETS assuming full recovery through auctions). By energy product, again diesel (2548.8 million euros) and natural gas (1470.9 million euros) would be the main sources of additional revenue (see Table 14).

**Table 13.** Reform 2B. Effects on energy products.

| Product | Price Variation | | Consumption Variation | | |
|---|---|---|---|---|---|
| | Residential | Industrial | Residential | Industrial | Total |
| Electricity | 2.11% | 3.99% | −0.43% | −0.81% | −0.70% |
| Natural Gas | 8.78% | 9.03% | −2.12% | −2.18% | −2.17% |
| Diesel | 11.49% | 11.49% | −2.31% | −2.31% | −2.31% |
| Gasoline | 8.28% | - | −2.09% | - | −2.09% |

Source: The authors.

**Table 14.** Reform 2B. Revenue changes. Millions of euros.

| | | Excise Tax | VAT | CO₂ Tax | NOx/SO₂ Tax | Renewables | Total | Total (%) |
|---|---|---|---|---|---|---|---|---|
| | Residential | 6.59 | 50.02 | 186.96 | 102.67 | −16.67 | 329.58 | 4.35 |
| Electricity | Industrial | 15.16 | - | 430.63 | 236.48 | −20.35 | 661.91 | 19.72 |
| | Total | 21.75 | 50.02 | 617.59 | 339.15 | −37.02 | 991.49 | 9.07 |
| | Residential | −2.84 | 51.43 | 305.22 | 20.14 | - | 373.95 | 40.26 |
| Natural Gas | Industrial | −2.28 | - | 1031.18 | 68.03 | - | 1096.93 | 1051.49 |
| | Total | −5.12 | 51.43 | 1336.41 | 88.17 | - | 1470.88 | 142.37 |
| | Residential | −145.07 | 267.79 | 1251.65 | 353.90 | - | 1728.27 | 18.62 |
| Diesel | Industrial | −81.50 | - | 703.18 | 198.82 | - | 820.51 | 23.26 |
| | Total | −226.57 | 267.79 | 1954.83 | 552.72 | - | 2548.78 | 19.90 |
| | Residential | -52.96 | 65.70 | 352.95 | 68.92 | - | 434.61 | 12.00 |
| Gasoline | Industrial | - | - | - | - | - | - | - |
| | Total | −52.96 | 65.70 | 352.95 | 68.92 | - | 434.61 | 12.00 |
| | **Residential** | **−194.28** | **434.94** | **2096.78** | **545.63** | **−16.67** | **2866.41** | **13.39** |
| **Total** | **Industrial** | **−68.62** | - | **2165.00** | **503.33** | **−20.35** | **2579.35** | **36.91** |
| | **Total** | **−262.90** | **434.94** | **4261.78** | **1048.96** | **−37.02** | **5445.76** | **19.18** |

Note: The last column depicts the change in revenue with respect to the baseline. Source: The authors.

Just like in the previous reform, the distributive effects fundamentally derive from the increase in the price of diesel, hence their similarity. However, the distributive effects of this reform are stronger (Figure 5), as there is a more intense decrease in the income level of all households until the eighth decile. By contrast, the slight decrease in the Gini index (0.01%) indicates that the reform is progressive.

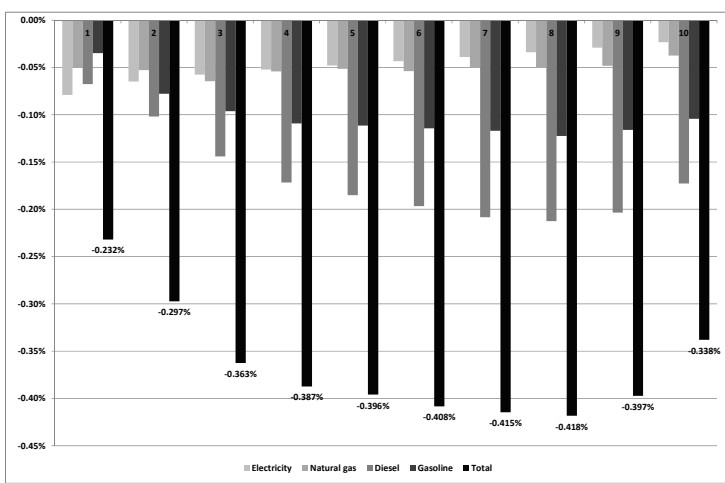

**Figure 5.** Reform 2B. Distributive impact by income deciles (%). Source: The authors.

*4.3. Scenario 3. Eliminating the Support costs for Renewable, Cogeneration and waste from the Electricity Bill*

4.3.1. Reform 3A. Eliminating the Cost of Supporting Renewable Energies, Cogeneration and Waste from the Electricity Tariff, through a Tax on the Energy Sectors (Consumption)

This reform considers eliminating the costs of supporting renewable energies, cogeneration and waste from the electricity tariff, financing these costs through a tax on all energy products and distributing the tax burden among energy products in proportion to their consumption, as explained in Section 3. Given the significant costs of promoting renewables (Section 3.1), this reform would cause a substantial reduction in the price of electricity whereas the prices of the remaining energy products would increase to finance part of the cost of renewables. This would result in increased electricity consumption and a reduced consumption of the remaining energy products of between 2.1% and 2.7% (Table 15), which would allow for a 0.54% reduction of $CO_2$ emissions associated with the consumption of these products.

**Table 15.** Reform 3A. Effects on energy products.

| Product | Price Variation | | Consumption Variation | | |
|---|---|---|---|---|---|
| | Residential | Industrial | Residential | Industrial | Total |
| Electricity | −18.86% | −9.33% | 3.83% | 1.89% | 2.48% |
| Natural Gas | 15.31% | 8.78% | −3.71% | −2.13% | −2.49% |
| Diesel | 12.57% | 7.24% | −2.53% | −1.46% | −2.14% |
| Gasoline | 10.70% | - | −2.71% | - | −2.71% |

Source: The authors.

The reform is revenue neutral (Table 16), so the new tax would allow for an additional revenue of 7000 million euros covering the cost of renewables, cogeneration and waste, as well as the fall in excise tax revenue (consequent to the reduction in energy consumption. (In the case of electricity, reduced excise tax revenue results from reducing the tax rate due to the fall in the price of electricity given its ad valorem nature, see Table 2) and VAT (the revenue provided by VAT is reduced in the case of electricity given the fall in spending while the revenue for the remaining energy products increases for the opposite reason; so the total VAT revenue hardly changes). By energy product, diesel (about 2400 million euros) and natural gas (1700 million euros) would provide the largest increase in tax revenue. In any case, electricity, after reducing the coverage of renewable costs, would increase tax revenues by about 1730 million euros.

**Table 16.** Reform 3A. Revenue changes. Millions of euros.

| | | Excise Tax | VAT | New Tax | Renewables | Total | Total (%) |
|---|---|---|---|---|---|---|---|
| Electricity | Residential | −180.05 | −471.00 | 1458.84 | −3891.26 | −3083.47 | −40.71 |
| | Industrial | −114.78 | - | 1035.43 | −2510.87 | −1590.22 | −47.37 |
| | Total | −294.83 | −471.00 | 2494.26 | −6402.13 | −4673.69 | −42.75 |
| Natural Gas | Residential | −4.96 | 87.78 | 558.29 | - | 641.12 | 69.03 |
| | Industrial | −2.22 | - | 1070.52 | - | 1068.30 | 1024.04 |
| | Total | −7.17 | 87.78 | 1628.82 | - | 1709.42 | 165.46 |
| Diesel | Residential | −158.59 | 291.94 | 1751.33 | - | 1884.68 | 20.31 |
| | Industrial | −51.35 | - | 573.26 | - | 521.92 | 14.80 |
| | Total | −209.94 | 291.94 | 2324.60 | - | 2406.60 | 18.79 |
| Gasoline | Residential | −68.46 | 84.21 | 541.92 | - | 557.67 | 15.40 |
| | Industrial | - | - | - | - | - | - |
| | Total | −68.46 | 84.21 | 541.92 | - | 557.67 | 15.40 |
| **Total** | **Residential** | **−412.06** | **−7.07** | **4310.39** | **−3891.26** | **0** | **0** |
| | **Industrial** | **−168.34** | **-** | **2679.21** | **−2510.87** | **0** | **0** |
| | **Total** | **−580.40** | **−7.07** | **6989.60** | **−6402.13** | **0** | **0** |

Note: The last column depicts the change in revenue with respect to the baseline. Source: The authors.

From a distributional point of view, this reform has a very progressive impact on households (Figure 6), that mainly derives from the reduction in the price of electricity and the increase in the price of diesel because they lead to increased household income level at the four poorest deciles and reduces that of the others, a decrease that becomes greater as the level of income increases up to the ninth decile. The Gini index also falls 0.26%.

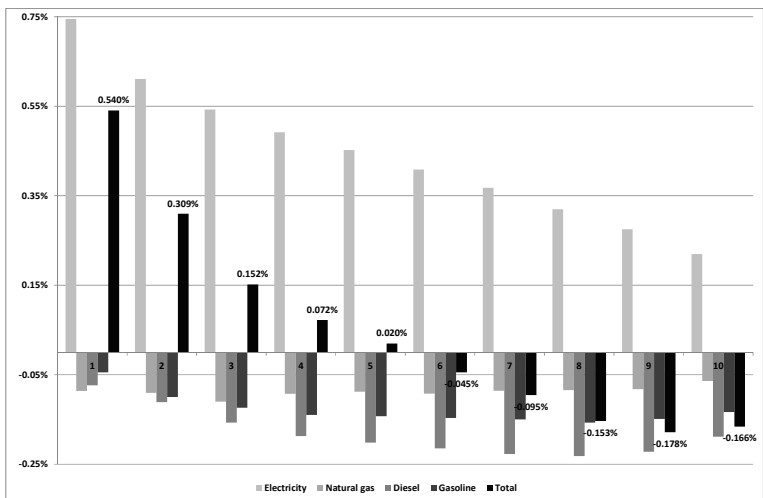

**Figure 6.** Reform 3A. Distributive impact by income deciles (%). Source: The authors.

4.3.2. Reform 3B. Suppression of the Costs of Supporting Renewable Energies, Cogeneration from Electricity Consumers and Financing these Costs with a Tax on the Energy Sectors that Considers Price Reaction (Ramsey)

This reform is similar to the previous one. However, as explained in Section 3.2, the distribution of the cost of financing renewables, cogeneration and waste among the energy products is inversely proportional to their price elasticity rather than being proportional to their consumption. Diesel becomes thus the product with a larger role in the coverage of renewable costs because it has the lowest price elasticity (in absolute value) (see Table 3), while gasoline finances a lower proportion of this cost. In any case, the share of residential natural gas consumption is lower than the proportion of the financed cost, so the residential price of this product will suffer the biggest increase (Table 17), while diesel will experience a smaller increase in its residential price due to the relevance of household consumption. The reduction in the residential price of electricity will be lower than in Reform 3A, given that the percent of the cost of renewables financed through electricity is larger than in Reform 3A. Consequently, the reduction in $CO_2$ emissions (0.51%) will be also slightly lower than in Reform 3A.

**Table 17.** Reform 3B. Effects on energy products.

| Product | Price Variation | | Consumption Variation | | |
|---|---|---|---|---|---|
| | Residential | Industrial | Residential | Industrial | Total |
| Electricity | −17.33% | −9.55% | 3.52% | 1.93% | 2.42% |
| Natural Gas | 21.72% | 6.04% | −5.26% | −1.46% | −2.33% |
| Diesel | 7.08% | 12.50% | −1.42% | −2.51% | −1.82% |
| Gasoline | 16.72% | - | −4.23% | - | −4.23% |

Source: The authors.

This reform is also revenue neutral (Table 18). The revenue obtained by the new tax will be slightly higher than in Reform 3A because it must compensate for a greater fall in excise tax and VAT revenues. By energy product, once again diesel (about 2000 million euros) and natural gas (approximately 1650 million euros) become the main sources of additional tax revenue.

**Table 18.** Reform 3B. Revenue changes. Millions of euros.

|  |  | Excise | VAT | New Tax | Renewables | Total | Total (%) |
|---|---|---|---|---|---|---|---|
| Electricity | Residential | −181.57 | −431.40 | 1678.73 | −3891.26 | −2825.50 | −37.30 |
|  | Industrial | −114.45 | - | 995.83 | −2510.87 | −1629.49 | −48.54 |
|  | Total | −296.02 | −431.40 | 2674.56 | −6402.13 | −4454.99 | −40.75 |
| Natural Gas | Residential | −7.03 | 121.84 | 779.20 | - | 894.00 | 96.25 |
|  | Industrial | −1.52 | - | 740.86 | - | 739.34 | 708.71 |
|  | Total | −8.55 | 121.84 | 1520.06 | - | 1633.34 | 158.10 |
| Diesel | Residential | −89.44 | 166.98 | 998.82 | - | 1076.36 | 11.60 |
|  | Industrial | −88.62 | - | 978.77 | - | 890.15 | 25.23 |
|  | Total | −178.06 | 166.98 | 1977.59 | - | 1966.51 | 15.35 |
| Gasoline | Residential | −106.95 | 128.77 | 833.31 | - | 855.13 | 23.62 |
|  | Industrial | - | - | - | - | - | - |
|  | Total | −106.95 | 128.77 | 833.31 | - | 855.13 | 23.62 |
| **Total** | **Residential** | **−384.99** | **−13.82** | **4290.06** | **−3891.26** | **0** | **0** |
|  | **Industrial** | **−204.59** | **-** | **2715.46** | **−2510.87** | **0** | **0** |
|  | **Total** | **−589.58** | **−13.82** | **7005.53** | **−6402.13** | **0** | **0** |

Note: The last column depicts the change in revenue with respect to the baseline. Source: The authors.

The distributive impacts (Figure 7) are akin to those of Reform 3A, with a slightly lower increase in income in the first four deciles (due to a smaller fall in the price of electricity) and a reduction in the level of income from the fifth decile forward. The impact on the Gini index is also slightly lower than in Reform 3A (−0.23%).

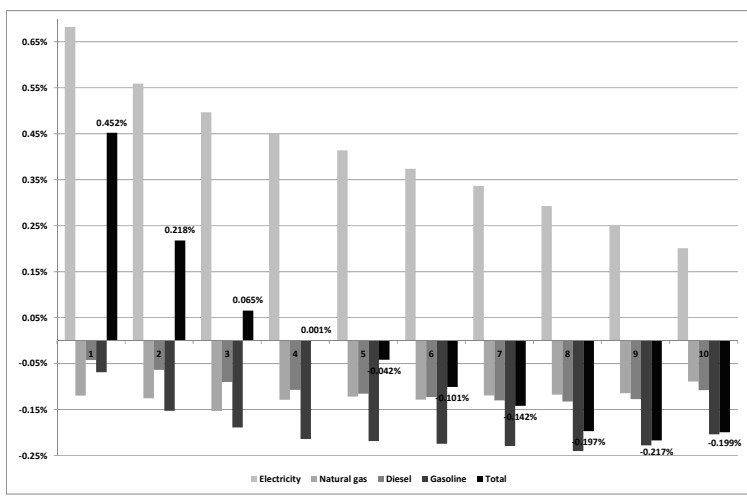

**Figure 7.** Reform 3B. Distributive impact by income deciles (%). Source: The authors.

*4.4. Scenario 4. Taxes on Emissions and the Financing of Renewables*

4.4.1. Reform 4A. Taxes on $CO_2$ Emissions (€30/t), $SO_2$ (€2000/t) and NOx (€2000/t) and the Suppression of the Cost of Promoting Renewables in the Electricity Bill

In this case, we consider introducing the same taxes on emissions as in Reform 2B, but also eliminate the costs of promoting renewables from the electricity bill, which represents a substantial part of final prices. Therefore, the impact of the reform on the price and consumption (and the associated tax revenue) of natural gas, diesel and gasoline would be the same as in simulation 2B, except for electricity, where prices would experience a significant fall (see Table 19) and thus would lead to a 3.2% increase in consumption, causing a lower reduction of $CO_2$ emissions associated with energy products than in Reform 2B (−0.21%).

**Table 19.** Reform 4A. Effects on energy products.

| Product | Price Variation | | Consumption Variation | | |
|---|---|---|---|---|---|
| | Residential | Industrial | Residential | Industrial | Total |
| Electricity | −26.61% | −11.18% | 5.40% | 2.26% | 3.22% |
| Natural gas | 8.78% | 9.03% | −2.12% | −2.18% | −2.17% |
| Diesel | 11.49% | 11.49% | −2.31% | −2.31% | −2.31% |
| Gasoline | 8.28% | - | −2.09% | - | −2.09% |

Source: The authors.

The reform could generate an additional 4500 million euros in tax revenue (Table 20). However, this amount would only partially cover the renewable, cogeneration and waste costs eliminated from the electricity bill (6400 million euros). The difference would therefore require funding outside of the energy sector. Hence, the revenue derived from taxes on emissions would be slightly higher than in the preceding simulation and the revenue from the VAT would fall as a result of the reduction in electricity price. Diesel would be the main source of additional revenue in this reform.

**Table 20.** Reform 4A. Revenue changes. Millions of euros.

| | | Excise Tax | VAT | $CO_2$ Tax | $NOx/SO_2$ Tax | Renewables | Total | Total (%) |
|---|---|---|---|---|---|---|---|---|
| Electricity | Residential | −162.16 | −677.28 | 197.91 | 108.68 | −3891.26 | −4424.10 | −58.41 |
| | Industrial | −89.39 | - | 444.00 | 243.82 | −2510.87 | −1912.43 | −56.97 |
| | Total | −251.54 | −677.28 | 641.91 | 352.50 | −6402.13 | −6336.54 | −57.97 |
| Natural gas | Residential | −2.84 | 51.43 | 305.22 | 20.14 | - | 373.95 | 40.26 |
| | Industrial | −2.28 | - | 1031.18 | 68.03 | - | 1096.93 | 1051.49 |
| | Total | −5.12 | 51.43 | 1336.41 | 88.17 | - | 1470.88 | 142.37 |
| Diesel | Residential | −145.07 | 267.79 | 1251.65 | 353.90 | - | 1728.27 | 18.62 |
| | Industrial | −81.50 | - | 703.18 | 198.82 | - | 820.51 | 23.26 |
| | Total | −226.57 | 267.79 | 1954.83 | 552.72 | - | 2548.78 | 19.90 |
| Gasoline | Residential | −52.96 | 65.70 | 352.95 | 68.92 | - | 434.61 | 12.00 |
| | Industrial | - | - | - | - | - | - | - |
| | Total | −52.96 | 65.70 | 352.95 | 68.92 | - | 434.61 | 12.00 |
| **Total** | **Residential** | **−363.03** | **−292.36** | **2107.73** | **551.64** | **−3891.26** | **−1887.27** | **−8.82** |
| | **Industrial** | **−173.16** | **-** | **2178.37** | **510.67** | **−2510.87** | **5.01** | **0.07** |
| | **Total** | **−536.19** | **−292.36** | **4286.10** | **1062.32** | **−6402.13** | **−1882.27** | **−6.63** |

Note: The last column depicts the change in revenue with respect to the baseline. Source: The authors.

The distributive impact of this reform on households would be very progressive, as a consequence of the significant fall in the price of electricity (Figure 8). The level of income of the households of all the deciles would increase, with the greater percentage increases coming from the lower levels of household income. The Gini index would decrease by 0.36%.

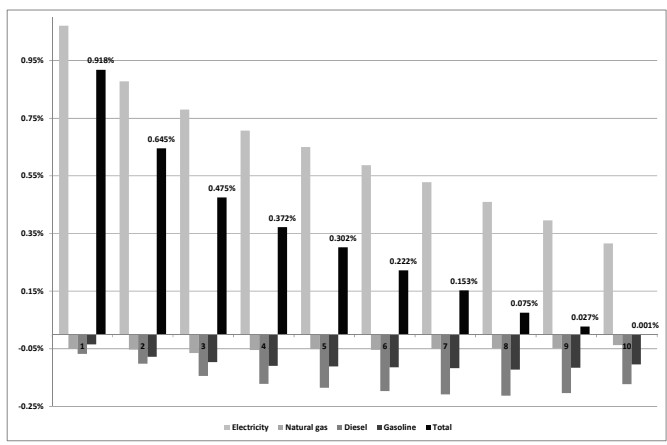

**Figure 8.** Reform 4A. Distributive impact by income deciles (%). Source: The authors.

4.4.2. Reform 4B. Taxes on $CO_2$, $SO_2$ and NOx Emissions and Suppression of the Cost of Promoting Renewable Electricity Rates. Revenue Neutrality

The reform considered in the previous section failed to generate enough revenue to achieve tax neutrality and caused a fall of almost 1900 million euros in tax revenues associated with energy products. This case therefore studies a modification of the tax rates on emissions that would lead to revenue neutrality.

To achieve this neutrality, the tax rates applied to emissions should increase by 32.1%, so that the tax rate on $CO_2$ emissions would be €39.6/t, while the tax rate of NOx and $SO_2$ would amount €2641.6/t. This would provoke a larger increase in the price of energy products than in Reform 4A, with a subsequent larger reduction in their consumption (Table 21), as well as in $CO_2$ emissions (0.79%). The revenue for financing the cost of supporting renewable, cogeneration and waste would be obtained from taxes on $CO_2$ (around 5200 million euros) and NOx and $SO_2$ (1439 million euros), which would also compensate for the reduction in the revenue of excise and VAT (Table 22). By energy product, diesel would continue to be the main source of additional revenue (over 3300 million euros) followed by natural gas (some 1900 million euros).

**Table 21.** Reform 4B. Effects on energy products.

| Product | Price Variation | | Consumption Variation | | |
|---------|-----------------|---|------------------------|---|---|
| | Residential | Industrial | Residential | Industrial | Total |
| Electricity | −25.62% | −9.30% | 5.20% | 1.89% | 2.89% |
| Natural gas | 11.60% | 11.92% | −2.81% | −2.88% | −2.87% |
| Diesel | 15.18% | 15.18% | −3.05% | −3.05% | −3.05% |
| Gasoline | 10.94% | - | −2.77% | - | −2.77% |

Source: The Authors.

**Table 22.** Reform 4B. Revenue change. Millions of euros.

| | | Excise Tax | VAT | $CO_2$ Tax | NOx/$SO_2$ Tax | Renewables | Total | Total (%) |
|---|---|------------|-----|-----------|----------------|------------|-------|-----------|
| | Residential | −158.31 | −650.53 | 292.59 | 156.95 | −3891.26 | −4250.56 | −56.12 |
| Electricity | Industrial | −81.32 | - | 655.21 | 351.47 | −2510.87 | −1585.51 | −47.23 |
| | Total | −239.63 | −650.53 | 947.80 | 508.42 | −6402.13 | −5836.07 | −53.39 |
| | Residential | −3.75 | 67.30 | 400.33 | 26.41 | - | 490.29 | 52.79 |
| Natural gas | Industrial | −3.01 | - | 1352.24 | 89.21 | - | 1438.44 | 1378.84 |
| | Total | −6.76 | 67.30 | 1752.57 | 115.62 | - | 1928.73 | 186.69 |
| | Residential | −191.61 | 350.32 | 1640.64 | 463.89 | - | 2263.25 | 24.38 |
| Diesel | Industrial | −107.65 | - | 921.72 | 260.61 | - | 1074.69 | 30.46 |
| | Total | −299.25 | 350.32 | 2562.37 | 724.50 | - | 3337.94 | 26.06 |
| | Residential | −69.95 | 85.97 | 462.97 | 90.41 | - | 569.41 | 15.73 |
| Gasoline | Industrial | - | - | - | - | - | - | - |
| | Total | −69.95 | 85.97 | 462.97 | 90.41 | - | 569.41 | 15.73 |
| | **Residential** | **−423.62** | **−146.93** | **2796.54** | **737.66** | **−3891.26** | **−927.62** | **−4.33** |
| **Total** | **Industrial** | **−191.97** | **-** | **2929.17** | **701.29** | **−2510.87** | **927.62** | **13.27** |
| | **Total** | **−615.60** | **−146.93** | **5725.71** | **1438.95** | **−6402.13** | **0** | **0** |

Note: The last column depicts the change in revenue with respect to the baseline. Source: The authors.

The distributive effect on households would be in this case progressive (Figure 9) since it increases the income level of households in the poorest deciles (up to the seventh decile). This increase is also greater in relative terms because the larger increase corresponds to the lower levels of income, and the income level of the richest deciles corresponds to a greater reduction (the higher the income, the higher the reduction). The Gini index falls by 0.36%, as in the previous simulation.

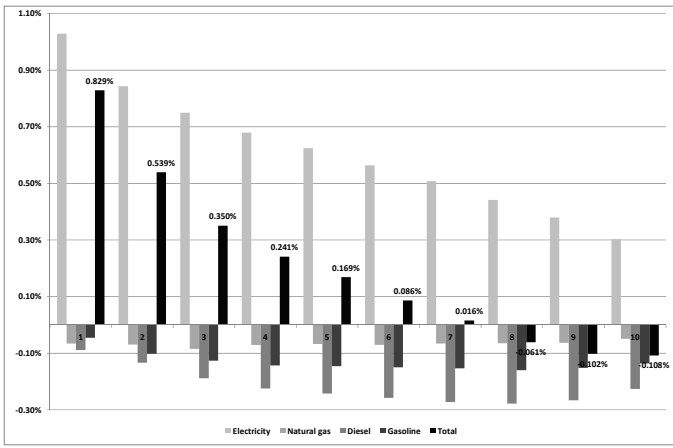

**Figure 9.** Reform 4B. Distributive impact by income deciles (%). Source: The authors.

## 5. Discussion

This article analyzes the results of a series of "third generation" green tax reforms in Spain. It explores their short-term effects on energy demand, $CO_2$ emissions and revenue, as well as the distributive impact on households. Table 23 summarizes the main results. In sum, the increase of the excises applied on energy products up to the average level in the main EU countries is the option that would allow for the largest increase in tax revenues. Likewise, the introduction of new taxes on emissions (especially if high tax rates are used) and the increase of excise taxes up to the minimum levels of the 2011 EU Directive proposal would also allow for an increase in tax revenue, although to a lesser extent. For its part, the combination of taxes on emissions and the elimination of the cost of renewable electricity rates would cause a fall in revenue (Reform 4A), which could be avoided by increasing the tax rates on emissions. Finally, the reforms of Scenario 3 are revenue neutral by definition.

Setting excise taxes on energy products on the average levels of the main EU countries would have the greatest impact on residential energy demand and $CO_2$ emissions, achieving a significant reduction in the demand for energy products and associated $CO_2$ emissions. The other reforms have a lower impact, but in all of the cases they reduce energy demand and emissions (It should be emphasized that we are carrying out a short-term analysis, without considering the long-term effects of substitution between energy products. In this sense, when there is a significant fall in the price of electricity together with an increase in the price of the remaining energy products (Scenarios 3 and 4), there will be a significant substitution effect in the long term, with the consumption of electricity increasing even more and vice versa for the other energy products. On the other hand, when there is a significant increase in the price of all energy products (Reforms 1B and 2B) and given the connection between energy consumption and aggregate production, in the long term there could be a contraction in GDP. However, there will be a simultaneous stimulus of clean energies, thus favoring the transition towards a low-carbon economy. In addition, these reforms could be complemented by other energy efficiency measures to achieve additional reductions in energy consumption and emissions (see, for example, [93]), thus mitigating the negative impacts on the economy).

Regarding the distributive impact on households, on average, only the reforms reducing the final price of electricity would increase household incomes. The impact of these reforms would be quite progressive because higher increases in income levels would largely correspond to lower household incomes. On the other hand, taxing emissions or raising excise taxes and allocating the revenue to fiscal consolidation would result in reduced household income, but the impact is slightly progressive given the increase in the price of diesel. Therefore, the reform raising the excise tax rate on energy products to the average level in European countries (Reform 1B) is the only one with a regressive impact that would, in fact, result from the significant increase in the price of electricity. In terms of the Gini index, all reforms except 1B reduce inequality.

Thus, in the face of a possible increase in EU energy taxation, in line with the demands of both the European Commission (see [9]) and several member states (see [94]) within the measures to fight climate change, raising Spanish energy taxes could lead to significant public revenues, while at the same time reducing energy demand and associated emissions, with a limited impact on households that is generally progressive. In addition, raising energy taxation would stimulate innovation and encourage the development of more energy-efficient alternative products and processes [95]. In any case, price increases in energy products can affect aggregate inflation. In this sense, Spain is in a moment of very low inflation and translation to the General Retail Price Index of the prices increases could represent a small contribution since the share of all energy goods considered in the basket of goods of the representative household was 7.2% in 2016. Of course, whenever translation of energy price increases to inflation become important, it can produce negative effects on competitiveness and growth.

**Table 23.** Summary of the impacts of the reforms.

| Reform | Income | Energy Demand | Emissions $CO_2$ | Distributive Impact (Deciles) | Gini Index |
|--------|--------|---------------|------------------|-------------------------------|------------|
| 1A | 5.97% | −0.65% | −0.55% | −0.041% (first)<br>−0.068% (tenth)<br>−0.071% (average) | −0.01% |
| 1B | 56.25% | −4.18% | −4.59% | −1.107% (first)<br>−0.899% (tenth)<br>−1.139% (average) | 0.18% |
| 2A | 6.11% | −0.58% | −0.53% | −0.064% (first)<br>−0.119% (tenth)<br>−0.122% (average) | −0.01% |
| 2B | 19.18% | −1.77% | −1.67% | −0.232% (first)<br>−0.338% (tenth)<br>−0.365% (average) | −0.01% |
| 3A | 0.00% | −0.91% | −0.54% | 0.540% (first)<br>−0.166% (tenth)<br>0.046% (average) | −0.26% |
| 3B | 0.00% | −0.87% | −0.51% | 0.452% (first)<br>−0.199% (tenth)<br>−0.016% (average | −0.23% |
| 4A | −6.63% | −0.61% | −0.21% | 0.918% (first)<br>0.001% (tenth)<br>0.319% (average) | −0.36% |
| 4B | 0.00% | −1.21% | −0.79% | 0.829% (first)<br>−0.108% (tenth)<br>0.196% (average) | −0.36% |

Source: The authors.

## 6. Conclusions

This article proposes a menu of possible reforms of energy-environmental taxation in Spain, thus facilitating decisions by policy makers, through the analysis of short-term impacts both in terms of revenues, energy demand, associated $CO_2$ emissions and, especially, in distributional terms. The results show that the proposed reforms would generate significant revenue, which could be used both for fiscal consolidation and to finance other climate-related policies, such as the promotion of renewables. In addition, the distributive impact of the reforms under consideration would, in general, be progressive, which obviously increases their viability and social acceptance. In summary, the environmental and socio-economic profile of the different simulations reflect the high potential of third generation green tax reforms in the case of Spain despite their scarce and imperfect use so far.

**Author Contributions:** All authors contributed equally to the paper.

**Funding:** This research received funding from the Spanish Ministry of Science.

**Acknowledgments:** The authors are thankful to four anonymous reviewers for their detailed comments and suggestions. General support from the Spanish Ministry of Science (project RTI2018-093692-B-I00, Xavier Labandeira and Xiral López-Otero and project ECO2015-70349-P, José M. Labeaga) is gratefully recognized. The usual disclaimer applies.

**Conflicts of Interest:** The authors declare no conflict of interest.

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
