# Peer review of "New Green Tax Reforms: Ex-Ante Assessments for Spain"

_sustainability, doi:10.3390/su11205640_

Round 1
Reviewer 1 Report
Dear authors,
The aim of the paper is interesting. It offers a simulation of different scenarios for changes in taxation on energy in Spain, with estimations on tax revenues and income redistribution. Although the paper is informative, it is methodologically naïve. The presentation and writing of the paper are clear.
There are two concerns about it that, if corrected, would improve the quality of the paper.
The first one is about the handle of elasticities of demand. It would be interesting to read more information about the estimations of elasticities. They are referenced from another publication, nevertheless, what’s the time span of these estimations: short/long term? Have these been estimated individually? (with constant prices of substitutes) or simultaneously for all the energy sources? Is there no information about cross-elasticities?
These considerations are quite important to compare different reforming scenarios with symmetric and asymmetric changes in the taxation of every source. For instance, Reform 4B would mean a substantial reduction in electricity prices, and increases in the rest of energy sources. We’d reasonable expect a significant substitution effect that has not been considered. Other reforms, like 2B, result in an increase in the cost of every source . . . and the estimations of demand variations are significantly negative for all sources. Given the connection energy consumption – aggregate production, . . . would this suppose a contraction of GDP?... or maybe this is a not reasonable application of demand
Second, connected to my last question, the discussion section should widely be extended to include other implications of every scenario. It is a relevant topic that of income distribution caused by different taxation schemes and, of course, the tax income. But the discussion about a taxation model on energy should make the connections to environmental directions from the European Union. So it should be put in the context of technological issues. Above all, regarding the economics approach of the paper, the authors should set the implications for aggregate inflation, competitiveness and growth. In this case, I do not mean the authors should perform new estimations, but extend the discussion section to provide the paper with a wider economic scope.
I hope my comments are useful to improve the quality of the paper.
Author Response
We would like to thank all your comments and suggestions. We have produced a new version of the article incorporating the comments and suggestions of the reviewers, which includes two new sections. On the one hand, we have divided the introductory section into two sections (introduction and theoretical context) and on the other hand we have added a section of conclusions. Specific answers to the reviewers' comments are provided below in separate documents

Reviewer 2 Report
Paper presents very important issue of socio-economic aspect of environmental taxes. Authors have done an ex-ante assessment of four scenarios of taxes implementation in Spain and analyze the predicted influence of taxes value on fuels prices, fuel consumption, CO2 emissions, total income from taxes and income of hauseholds. This article show the effects of taxes reform on socio-economic aspect and could be a good source material for legislators.
The article is very interesting and comprehensivle deals with the subject. Introduction is very well prepared, the same methodology. Results are clearly presented and discussed.
My only concern is that there is a lack of conclusion or the summary section of the article. It will be good if Authors make some short conclusion section to summ up the achievements of the article, maybe also with recomendation of scenario to use by legislators.

Author Response
We would like to thank all your comments and suggestions.
We have produced a new version of the article incorporating all your comments and suggestions, which includes two new sections. On the one hand, we have divided the introductory section into two sections (introduction and theoretical context) and on the other hand we have added a section of conclusions. Specific answers are provided below in separate files.

Reviewer 3 Report
Dear authors, please see the following for my comments
The introduction is quite complex and should be separated to have an introductory section and theoretical section. In the methods (section 2), can the source of elasticities in Table 3be better described? The Reference (i.e. Ref 67, Labandeira et al (2016) is not in English and not easy to translate). The methodology should be properly described to ensure replication of the present study. What are the effects of tax reforms on technological change in the energy mix? The authors should provide more extended discussion and a conclusion section
Author Response

(The authors gave the same response as above.)

Reviewer 4 Report
Dear Authors,
First of all, I would like to tell you that the work you have done is very interesting and at the same time it is important that the contents of the work are known. I believe that Spain urgently needs to reform its tax system so that it can be an instrument to help make the transition to a more sustainable and low-carbon economy.
I think that your research should be reinforced with some quotes and ideas to strengthen your hypotheses and discussion, but above all so that it can change the opinion of policy makers so that they can create a new fiscal model that is more effective in terms of sustainability and that helps to eliminate the negative externalities of the linear economy model.
I believe that in the discussion part, more ideas should be put forward so that public decision-makers are aware of the importance of fiscal proposals in environmental matters.
Proposals:
Keywords: ODS ONU, negative externalities, new generation green tax reforms in Spain, climate change mitigation, energy tax
Introduction
He, P.; Sun, Y.; Shen, H.; Jian, J.; Yu, Z. Does Environmental Tax Affect Energy Efficiency? An Empirical Study of Energy Efficiency in OECD Countries Based on DEA and Logit Model. Sustainability 2019, 11, 3792.
Villar Rubio, E., & Quesada Rubio, J., & Molina Moreno, V. (2017). ENVIRONMENTAL FISCAL EFFORT: SPATIAL CONVERGENCE WITHIN ECONOMIC POLICY ON TAXATION. Revista de Economía Mundial, (45), 87-100.
Kondo, R.; Kinoshita, Y.; Yamada, T. Green Procurement Decisions with Carbon Leakage by Global Suppliers and Order Quantities under Different Carbon Tax. Sustainability 2019, 11, 3710.
Tsai, W.-H. Carbon Taxes and Carbon Right Costs Analysis for the Tire Industry. Energies 2018, 11, 2121.
SAGE Publications UK. (2011, January 5). Carbon taxes are the answer to stalled climate negotiations, expert says. ScienceDaily. Retrieved June 11, 2019 from www.sciencedaily.com/releases/2011/01/110105194840.htm
Methodology and material
Liobikienė, G.; Butkus, M.; Matuzevičiūtė, K. The Contribution of Energy Taxes to Climate Change Policy in the European Union (EU). Resources 2019, 8, 63.
Villar Rubio, E.; QuesadaRubio, J.M.; Molina Moreno, V. CONVERGENCE ANALYSIS OF ENVIRONMENTAL FISCAL PRESSURE ACROSS EU-15 COUNTRIES ENERGY AND ENVIRONMENT 2015, 26 (5): 789-802, ISSN: 0958-305X
SAGE Publications UK. (2011, January 5). Carbon taxes are the answer to stalled climate negotiations, expert says. ScienceDaily. Retrieved June 11, 2019 from www.sciencedaily.com/releases/2011/01/110105194840.htm
Discussion
Streimikiene, D.; Siksnelyte, I.; Zavadskas, E.K.; Cavallaro, F. The Impact of Greening Tax Systems on Sustainable Energy Development in the Baltic States. Energies 2018, 11, 1193.
Molina-Moreno, V.; Núñez-Cacho Utrilla, P.; Cortés-García, F.J.; Peña-García, A. The Use of Led Technology and Biomass to Power Public Lighting in a Local Context: The Case of Baeza (Spain). Energies 2018, 11, 1783.
Author Response

(The authors gave the same response as above.)

Round 2
Reviewer 1 Report
The authors have answered to some of the concerns of this reviewer. They did not make major changes to the paper, but they included some comments and additional explanations that make explicit the limitations of the research and their results.
I still miss some more explanations about economic and industrial policy implications that would have make the paper more informative and valuable.
Anyway, the paper in its present form meets the standards of this journal. Therefore, my recommendation to the editor is to accept.
Reviewer 4 Report
Dear Authors. In the first place, I want to congratulate you for having incorporated into your article the most important recommendations that I had made to you and that had as an aim to improve the quality and subsequent dissemination of this research. On the other hand, I also want to motivate you so that your research team continues in this important field of research.